# HiFi-123: Towards High-fidelity One Image to 3D Content Generation

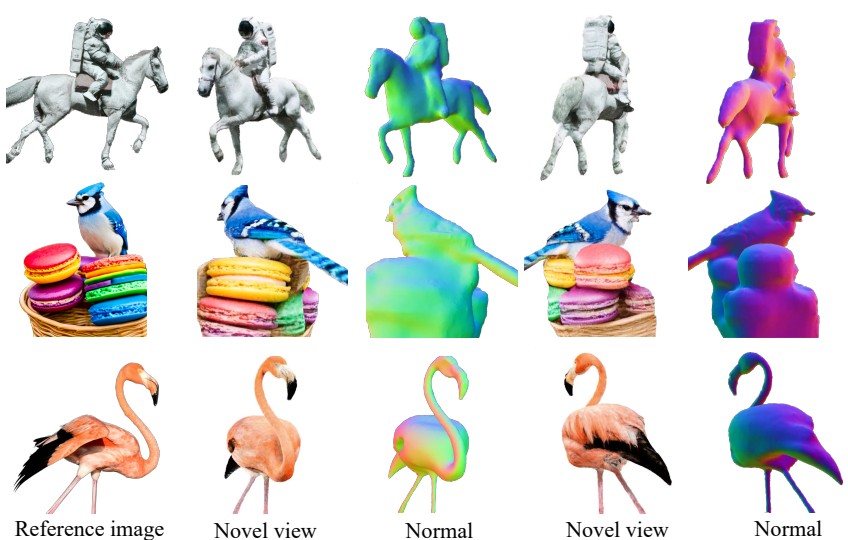

| Reference image | Novel view | Normal | Novel view | Normal |

Figure 1: *HiFi-123* is capable of generating high-fidelity 3D content from a single reference image. The presented novel views and normal maps of the generated 3D content demonstrate that our approach maintains consistency with the reference image, even in views significantly deviating from the reference view.

## Abstract

Recent advances in text-to-image diffusion models have enabled 3D generation from a single image. However, current image-to-3D methods often produce suboptimal results for novel views, with blurred textures and deviations from the reference image, limiting their practical applications. In this paper, we introduce *HiFi-123*, a method designed for high-fidelity and multi-view consistent 3D generation. Our contributions are twofold: First, we propose a reference-guided novel view enhancement technique that substantially reduces the quality gap between synthesized and reference views. Second, capitalizing on the novel view enhancement, we present a novel reference-guided state distillation loss. When incorporated into the optimization-based image-to-3D pipeline, our method significantly improves 3D generation quality, achieving state-of-the-art performance. Comprehensive evaluations demonstrate the effectiveness of our approach over existing methods, both qualitatively and quantitatively. Video comparisons are available on the supplementary project page. We will release our code to the public.

## 1 Introduction

The generation of 3D digital content is a fundamental task in computer vision and computer graphics with applications in robotics, virtual reality, and augmented reality. Producing such 3D content often demands proficiency in specialized software tools, setting a high threshold in terms of skill and cost. An alternative approach is through 3D digitization, which often relies on a large set of multi-view

images and their corresponding camera poses; However, acquiring such data is challenging. A more ambitious approach is to construct 3D content from only a single image, whether obtained from the web or generated. While humans can intuitively infer 3D shapes and textures from 2D images, creating 3D assets from a single image using computer vision techniques is difficult due to the limited 3D cues and ambiguities of a single viewpoint.

Recent advances in diffusion models, trained on web-scale 2D image datasets, have led to significant improvements in image generation (Ramesh et al., 2021; Rombach et al., 2022; Balaji et al., 2022; Nichol et al., 2021) and have been extended to video synthesis (Khachatryan et al., 2023; Zhang et al., 2023). These models, capable of generating high-quality and diverse content, offer controllability (Zhang & Agrawala, 2023; Mou et al., 2023) and cross-modal generation capabilities, such as text-to-image synthesis (Ramesh et al., 2021; Rombach et al., 2022; Balaji et al., 2022) (T2I). By leveraging the 3D priors inherent in T2I models, methods such as (Poole et al., 2022; Wang et al., 2023b) have utilized score distillation sampling (SDS) to achieve notable results in text-to-3D creation. This progress has also influenced the image-to-3D domain, with works such as (Melas-Kyriazi et al., 2023; Xu et al., 2023; Tang et al., 2023a) employing SDS loss combined with reference-view pixel losses to optimize neural representations from a single image. While these methods can produce reasonable 3D structures from a single viewpoint, the visual quality in *novel views* often *lacks fidelity*, exhibiting *inconsistencies* with the reference view as shown in Fig. 5. These challenges not only compromise the realism of the generated content but also limit its potential for broader applications.

The primary challenge arises when, for novel views outside the reference view, the optimization becomes overly reliant on, and thus closely tied to, the (inferred) text prompt. These text prompts, even those derived through textual inversion (Gal et al., 2022a; Burgess et al., 2023), often fail to capture the full visual details of the reference view, leading to inconsistent optimization results. The strong CFG guidance present in the SDS loss further amplifies this issue. Recently, Zero-1-to-3 (Liu et al., 2023b) introduced an approach that demonstrated the efficacy of large-scale diffusion models for zero-shot novel view synthesis, highlighting their ability to produce views consistent with the input. Following this direction, subsequent studies (Qian et al., 2023; Shi et al., 2023) integrated this "multi-view-aware" capability into the SDS optimization framework, achieving improved consistency in image-to-3D. However, models like Zero-1-to-3 require fine-tuning on (synthetic) multi-view datasets, which can lead to a noticeable degradation in model performance, particularly in generating unnatural and low-quality images in novel views.

In this work, we aim to devise a method that simultaneously generates consistent novel views from a single image while maintaining high image quality. Our primary insight lies in the application of the image inversion technique (Song et al., 2020) to retain the detailed structure and content of a specific image, enabling the generation of novel views and the subsequent 3D representation of the object. We observe that by integrating depth information into the DDIM inversion (Song et al., 2020) and the sampling process, based on a depth-conditioned diffusion model (SDD), the reconstruction quality is significantly improved. Leveraging this observation, we introduce *HiFi-123*, a method that, while intuitive, effectively generates high-fidelity 3D content from a single input image. We design a dual-branch inversion scheme in which both the reference image and a "coarse" estimation of the target view are inverted and reconstructed concurrently. A key feature of our approach is the use of the attention injection technique (Wu et al., 2022; Qi et al., 2023; Cao et al., 2023) during the generation phase to transfer the appearance of the target object from the reference view to the coarse novel view, a process we term "novel view enhancement." This enhancement mechanism can be seamlessly integrated into the recent coarse-to-fine 3D generation framework (Lin et al., 2023). Moreover, the inversion process's unique properties enable us to re-formulate and re-derive the SDS loss, resulting in a Reference-Guided State Distillation loss that is intuitive and easy to implement. As a result, we can achieve high-fidelity image-to-3D generation that significantly exceeds prior techniques.

We comprehensively evaluated HiFi-123 on both zero-shot novel view synthesis and image-to-3D generation tasks. Both qualitative and quantitative results indicate that our approach excels in generating high-fidelity and consistent novel views from a single input image and further produces high-quality 3D content. Compared to state-of-the-art approaches, our method shows significant improvements in visual quality, contextual distance, and CLIP-similarity, marking an important step towards more accessible and democratized 3D content creation.

In summary, the main contributions of our work are two-fold:

- We introduce a reference-guided novel view enhancement method grounded in the inversion process. This approach can function as a plug-and-play module to improve the fidelity of results derived from diffusion-based novel view synthesis methods.

- Leveraging the aforementioned novel view enhancement, we present a new Reference-Guided State Distillation loss. When incorporated into the optimization-based image-to-3D framework, it significantly enhances the quality of 3D generation, achieving state-of-the-art performance.

## 2 RELATED WORK

### 2.1 DIFFUSION MODELS

Recently, diffusion models (Sohl-Dickstein et al., 2015; Ho et al., 2020) have drawn lots of attention due to their high-quality generation results that are comparable to previous generative adversarial networks (Goodfellow et al., 2014), which usually require specific optimization strategies to stabilize training. However, diffusion models usually need many iterations of denoising to produce a sample. To accelerate this process, denoising diffusion implicit models (DDIMs) (Song et al., 2020) propose to use non-Markovian diffusion processes which lead to short generative Markov chains with small generation steps. (Dhariwal & Nichol, 2021) propose classifier guidance to improve the sample quality of a diffusion model using an additional trained classifier that could trade off diversity for fidelity. While (Ho & Salimans, 2022) shows that the guidance can be performed by a pure generative model without the classifier by mixing the score estimates of a conditional diffusion model and a jointly trained unconditional diffusion model. Based on the diffusion models and large-scale image-text datasets, image synthesis conditioned on text prompt has achieved impressive results such as Glide (Nichol et al., 2021), DALL-E 2 (Ramesh et al., 2022), Imagen (Saharia et al., 2022) and StableDiffusion (Rombach et al., 2022). Recently, these pre-trained 2D text-to-image diffusion models have been taken as the image prior for other tasks like image editing and 3D generation and achieved high-quality results.

### 2.2 3D GENERATION USING 2D DIFFUSION

Based on the powerful text-to-image diffusion models in recent years, text-to-3D generation has also made great progress. DreamField (Jain et al., 2022) uses aligned image and text models to optimize NeRF (Mildenhall et al., 2021) without 3D shape or multi-view data. DreamFusion (Poole et al., 2022) proposes a Score Distillation Sampling (SDS) method that replaces CLIP loss from DreamField with a loss derived from the distillation of a 2D diffusion model to optimize a parametric NeRF model, which becomes a paradigm for 3D generation using 2D diffusion. A similar idea has also been presented in Score Jacobian Chaining (Wang et al., 2023a), which is a concurrent work of DreamFusion. Magic3D (Lin et al., 2023) builds upon DreamFusion that introduces several design choices like coarse-to-fine optimization, using Instant NGP representation in the coarse stage and 3D mesh representation in the fine stage. While Latent-NeRF Metzer et al. (2023) uses NeRF representation in latent space as the 3D model and Stable Diffusion as the 2D diffusion prior. To improve the text-to-3D generation results, Fantasia3D Chen et al. (2023a) disentangles the modeling of geometry and appearance, and ProlificDreamer (Wang et al., 2023b) proposes to modify score distillation sampling to variational score distillation which models the 3D parameters as a random variable instead of a constant. Most recently, many other works like HiFA (Zhu & Zhuang, 2023), IT3D (Chen et al., 2023b), HD-Fusion (Wu et al., 2023), EfficientDreamer (Zhao et al., 2023), have also been proposed to boost the text-to-3D generation.

Apart from text-to-3D generation, 3D generation based on a single image using diffusion models (image-to-3D) has also made rapid progress. NeuralLift-360 (Xu et al., 2023) learns to recover a 3D object from a single reference image with CLIP-guided diffusion prior. In addition to using the SDS loss for distillation, RealFusion (Melas-Kyriazi et al., 2023) and NeRDi (Deng et al., 2023) also adopt textual inversion to condition the diffusion model on a prompt with a token inverted by the reference image. While Make-It-3D (Tang et al., 2023a) employs textured point clouds as the representation in the fine stage to achieve high-quality results and Magic123 (Qian et al., 2023)

suggests using an additional 3D diffusion prior trained on large-scale multi-view dataset for score distillation sampling.

## 2.3 Multi-view Diffusion

Trained on large-scale 2D image datasets, the 2D text-to-image diffusion models could generalize to unseen scenes and different viewing angles that could be used for distilling 3D assets. However, due to the data bias of 2D images, e.g., most images are captured from front views, the 2D diffusion model may lack multi-view knowledge for 3D generation. Some efforts have been made to train the diffusion with 3D awareness. 3DiM (Watson et al., 2022) and Zero-1-to-3 (Liu et al., 2023b) present viewpoint-conditioned diffusion model for novel view synthesis trained on multi-view images. Utilizing large-scale 3D data, Zero-1-to-3 achieves zero-shot generalization ability to unseen images. One-2-3-45 (Liu et al., 2023a) uses the model from Zero-1-to-3 to generate multi-view images from the input view and leverage the generated results for 3D reconstruction. The recently arXived works (Shi et al., 2023; Burgess et al., 2023; Tang et al., 2023b; Liu et al., 2023c) try to generate multiview-consistent images from a single-view (Liu et al., 2023c) or text prompt (Shi et al., 2023). MVDream (Shi et al., 2023) leverages image diffusion models and a multi-view dataset to train a multi-view diffusion model, while SyncDreamer (Liu et al., 2023c) synchronizes the intermediate states of all the generated multi-view images at every step of the reverse process through a 3D-aware feature attention mechanism. However, these methods usually produce lower-quality results compared with the input view. In this paper, we propose a reference-based novel-view enhancement method based on DDIM inversion that could maintain the texture details in the novel view. Using this strategy, we also show the possibility of optimizing high-quality 3D content with our generation method.

## 3 Methodology

### 3.1 Preliminary

**Diffusion models.** A diffusion model consists of a forward process $q$ and a reverse process $p$. In the forward process, starting from a clean data $\boldsymbol{x}_0 \sim q_0(\boldsymbol{x}_0)$, noise is gradually added to the data point $\boldsymbol{x}_0$ to construct noisy state at different time steps. A notable property of the forward process is that noisy state $\boldsymbol{x}_t$ at time step $t$ could be formulated in the close form: $\boldsymbol{x}_t = \alpha_t \boldsymbol{x}_0 + \sigma_t \epsilon$, where $\alpha_t$ and $\sigma_t$ are hyper-parameters satisfying $\alpha_t^2 + \sigma_t^2 = 1$, $\epsilon \sim \mathcal{N}(\mathbf{0}, \boldsymbol{I})$. This results in a smoothed data distribution $q(\boldsymbol{x}_t) = \int q(\boldsymbol{x}_t|\boldsymbol{x}_0)q(\boldsymbol{x}_0)d\boldsymbol{x}_0$, where $q(\boldsymbol{x}_t|\boldsymbol{x}_0) = \mathcal{N}(\alpha_t \boldsymbol{x}_0, \sigma_t^2 \boldsymbol{I})$. The reverse process $p_\phi$ is defined by removing noise added on the clean data using a U-Net noise predictor $\epsilon_\phi$. In text-to-image diffusion models (Saharia et al., 2022; Ramesh et al., 2022; Rombach et al., 2022), $\epsilon_\phi$ is trained by minimizing the score matching objective:

$$\mathcal{L}_{\text{Diff}}(\phi) = \mathbb{E}_{\boldsymbol{x}_0 \sim q_0(\boldsymbol{x}_0), t \sim \mathcal{U}(0,1), \epsilon \sim \mathcal{N}(\mathbf{0}, \boldsymbol{I})}[w(t)\|\epsilon_\phi(\boldsymbol{x}_t; y, t) - \epsilon\|_2^2], \quad (1)$$

where $w(t)$ is a time-dependent weighting function and $y$ is conditional text embedding. After training, the predicted $\epsilon_\phi$ can be related to the score function of $p_\phi$, expressed as $\epsilon_\phi(\boldsymbol{x}_t; y, t) = -\sigma_t s_\phi(\boldsymbol{x}_t; y, t)$. To balance the quality and diversity of the generated images, classifier-free guidance (CFG Ho & Salimans (2022)) is adopted to modify the estimated noise as a combination of conditional and unconditional output: $\hat{\epsilon}_\phi(\boldsymbol{x}_t; y, t) = (1 + s)\epsilon_\phi(\boldsymbol{x}_t; y, t) - s\epsilon_\phi(\boldsymbol{x}_t; t)$, where $s > 0$ is the guidance scale. Increasing the guidance scale typically enhances the alignment between text and image, but at the cost of reduced diversity.

**DDIM inversion.** In the reverse process $p_\phi$, diffusion models often utilize deterministic DDIM sampling (Song et al., 2020) to speed up inference. DDIM sampling converts random noise $\boldsymbol{x}_T$ into clean data $\boldsymbol{x}_0$ over a sequence of discrete time steps, from $t = T$ to $t = 1$, formulated as: $\boldsymbol{x}_{t-1} = (\alpha_{t-1}/\alpha_t)(\boldsymbol{x}_t - \sigma_t \epsilon_\phi) + \sigma_{t-1} \epsilon_\phi$.

In contrast, DDIM inversion (Dhariwal & Nichol, 2021; Song et al., 2020) is a forward process that gradually converts a clean data $\boldsymbol{x}_0$ back to a noisy state $\boldsymbol{x}_T$ using denoising U-Net $\epsilon_\phi$. From $t = 1$ to $t = T$, we have $\boldsymbol{x}_t = (\alpha_t/\alpha_{t-1})(\boldsymbol{x}_{t-1} - \sigma_{t-1}\epsilon_\phi) + \sigma_t \epsilon_\phi$. In the case of unconditional generation, the DDIM inversion process $q_\phi$ is completely consistent with the sampling process $p_\phi$, so that the original data $\boldsymbol{x}_0$ can be precisely reconstructed by applying DDIM sampling on the inverted $\boldsymbol{x}_T$.

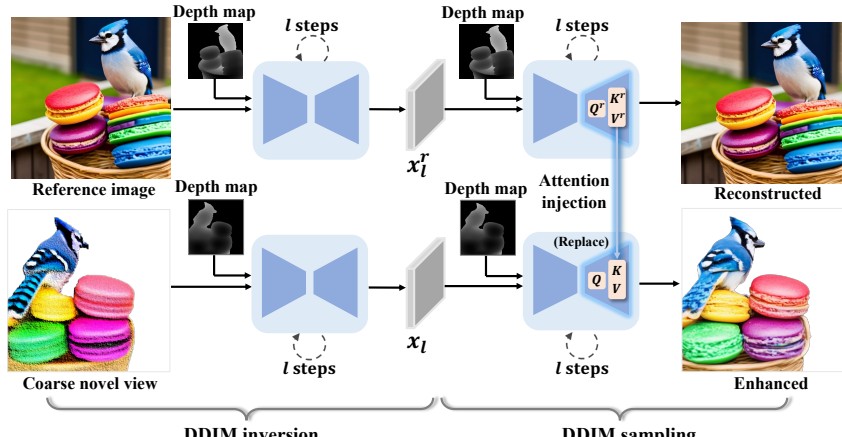

Figure 2: Reference-guided novel view enhancement pipeline. Our pipeline performs DDIM inversion and sampling on both the reference image and coarse novel view, and utilizes attention injection to transfer detail textures from the reference image to the coarse novel view.

However, for the text-conditioned generation with classifier-free guidance, the two processes are not consistent and the reconstruction quality will significantly decrease.

**Score distillation sampling (SDS).** SDS (Poole et al., 2022) is an optimization method commonly used in recent text-to-3D generation (Poole et al., 2022; Wang et al., 2023a; Lin et al., 2023; Metzer et al., 2023; Wang et al., 2023a; Chen et al., 2023a) and image-to-3D generation methods (Melas-Kyriazi et al., 2023; Qian et al., 2023; Xu et al., 2023; Tang et al., 2023a). The core idea of SDS is to distill prior knowledge from pre-trained text-to-image diffusion models to optimize the parameters of the 3D representation by minimizing:

$$\mathcal{L}_{\text{SDS}}(\theta) = \mathbb{E}_t \left[ (\sigma_t/\alpha_t)w(t)\text{KL}(q(\boldsymbol{x}_t|g(\theta, c); y, t)\|p_\phi(\boldsymbol{x}_t; y, t)) \right], \quad (2)$$

where $\theta$ denotes the parameters of a trainable 3D representation (e.g., NeRF (Mildenhall et al., 2021) or DMTet (Shen et al., 2021b)) and $g(\theta, c)$ is a rendered image given a camera pose $c$. By minimizing the KL divergence between distributions of noisy renderings and denoised images at different time steps, the 3D representation will be optimized to match the distribution of the images synthesized by the text-to-image diffusion model.

In practice, the gradients of Eq. 2 is approximated by:

$$\nabla_\theta \mathcal{L}_{\text{SDS}}(\theta) \approx \mathbb{E}_{t,\epsilon} \left[ w(t) \left( \epsilon_\phi(\boldsymbol{x}_t; y, t) - \epsilon \right) \frac{\partial \boldsymbol{x}_0}{\partial \theta} \right]. \quad (3)$$

Although optimizing with SDS loss can result in overall reasonable geometry, the generated 3D model often exhibits over-saturated colors and over-smoothed textures (Poole et al., 2022), which could lead to inconsistent results compared with the reference image when applied to image-to-3D generation tasks.

### 3.2 REFERENCE-GUIDED NOVEL VIEW ENHANCEMENT

Given a reference image, previous novel view synthesis methods based on diffusion models (Liu et al., 2023b; Poole et al., 2022) usually produce degraded and inconsistent results in novel views compared with the reference view. To tackle this problem, we propose a reference-guided novel view enhancement pipeline to transfer the detailed textures of the reference image to the coarse novel view. Our pipeline is built upon a discovery that incorporates a depth map into the DDIM inversion and sampling process using a depth-conditioned diffusion model (SDD) will significantly improve the reconstruction quality of the input image, as shown in Fig. 6. With this discovery, we can obtain the initial noise and the reverse processes that can faithfully reconstruct the detailed textures of the reference image. Then, inspired by the progressive generation property of the reverse process where the geometry structure emerges first at the early denoising steps while texture details

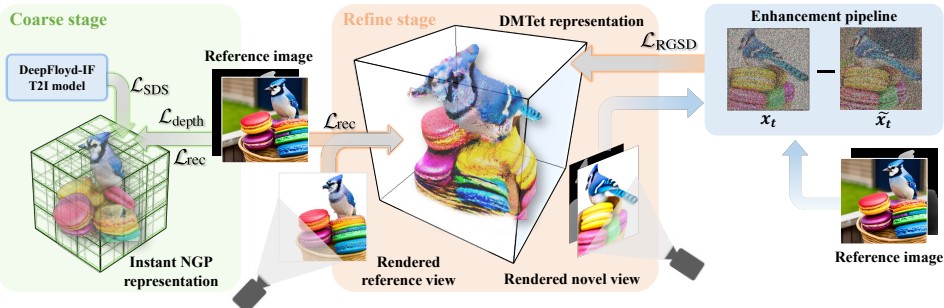

Figure 3: Image-to-3D generation pipeline. We utilize two stages to generate high-fidelity 3D contents. In the coarse stage, we optimize an Instant-NGP representation using SDS loss, reference view reconstruction loss and depth loss. In the refine stage, we export DMTet representation and use our proposed novel view enhancement pipeline to supervise training.

appear at the late denoising steps, we design a dual-branch pipeline to transfer fine textures of the reference image to the coarse novel views.

Specifically, our pipeline performs DDIM inversion and sampling on both the reference image and coarse novel view. In the forward process $q_\phi$, we separately map the reference image and coarse novel view back to the initial noisy state $\boldsymbol{x}_T^r$ and $\boldsymbol{x}_T$, with $t = T$ steps' DDIM inversion. In the reverse process, we first perform DDIM sampling separately on the two states to denoise them to the same intermediate noisy state indexed by $t = T - l$, where geometry structure has emerged yet textures are not. Then, in the following $t = l$ denoising steps where fine textures will gradually appear in the reference branch, we utilize an attention injection method (Wu et al., 2022; Qi et al., 2023; Cao et al., 2023) to inject attention features from the reference branch to the coarse novel view branch. In this way, fine textures of the reference image will be transferred to the coarse novel view. Thanks to the nearly perfect reconstruction quality of depth-based DDIM inversion, the inversion process and the sampling process are nearly consistent at every time step, we can thus simplify the pipeline into Fig. 2, where we directly invert the two inputs for $t = l$ steps, and then symmetrically adopt $t = l$ denoising steps with attention injection to propagate textures of the reference view to the coarse novel view.

Here, we denote the reverse process of the enhancement pipeline as $\tilde{p}_\phi$, as it requires attention injection, which is different from the regular reverse process $p_\phi$ of text-to-image generation. This pipeline can serve as a plug-and-play method for enhancing the quality of zero-shot novel view synthesis methods (Liu et al., 2023b;c), as shown in Fig. 4 and Fig. 12. We also demonstrated it can improve image-to-3D generation in the following section.

### 3.3 HIGH-FIDELITY 3D CONTENT GENERATION

Similar to recent text/image-to-3D generation approaches (Lin et al., 2023; Qian et al., 2023; Chen et al., 2023a; Tang et al., 2023a), we adopt a coarse-to-fine optimization strategy to create 3D contents. As shown in Fig. 3, in the coarse stage, we use SDS loss provided by an image space diffusion model (DF) along with reference view reconstruction loss and depth loss to optimize a coarse NeRF (Mildenhall et al., 2021; Müller et al., 2022). In the refine stage, we convert the implicit NeRF into an explicit mesh representation (Shen et al., 2021a) for higher rendering resolution and efficient training. Existing approaches (Tang et al., 2023a; Qian et al., 2023) continue optimizing the 3D model using SDS loss in the refine stage. Nonetheless, it can be observed in Eq. 2 that SDS loss leads to an optimization direction that forces the forward process $q$ of rendered novel views to approach the distribution of the reverse process $p_\phi$ of text-to-image generation. Due to the ambiguity of the inferred text descriptions and the large CFG guidance, the optimized novel views are often inconsistent with the reference image, as shown in Fig. 9. (a). To ensure high fidelity in novel views, we incorporate our proposed reference-guided novel view enhancement pipeline into the refine stage. Specifically, Inspired by the original formulation of SDS loss (Eq. 2), we construct a series of optimization targets using noisy states from the latent space of the enhancement pipeline.

The resulting new objective can be formulated as:

$$\min_{\theta} \mathbb{E}_{t \sim \mathcal{U}(0, l/T)}[\text{KL}(q_{\phi}(\boldsymbol{x}_t | g(\theta, c); y, m, t) \| \tilde{p}_{\phi}(\boldsymbol{x}_t; y, m, t))], \tag{4}$$

where $m$ denotes the conditioned depth map. Compared with Eq. 2, this improved objective forces noisy states of the rendered novel views to approach the enhanced states produced by the reference-guided novel view enhancement pipeline, ensuring that the supervision from the reference image can cover all the camera views, thereby endowing an accurate optimization direction towards the distribution of the 3D object that is consistent with the reference image. Since $q_{\phi}$ and $\tilde{p}_{\phi}$ are deterministic processes given a specific reference image, we can solve Eq. 4 using a distance metric (Huang et al., 2020) such as L2 distance. In this way, Eq. 4 can be simplified as:

$$\min_{\theta} \mathbb{E}_{t \sim \mathcal{U}(0, l/T)}[\|\boldsymbol{x}_t - \tilde{\boldsymbol{x}}_t\|_2^2], \tag{5}$$

where the noisy states $\boldsymbol{x}_t$ and the enhanced states $\tilde{\boldsymbol{x}}_t$ are obtained using the enhancement pipeline depicted in Fig. 2. Specifically, we invert the rendered novel view $\boldsymbol{x}_0 = g(\theta, c)$ into a noisy state $\boldsymbol{x}_l \sim q_{\phi}(\boldsymbol{x}_l)$ with $l$-step DDIM inversion, then perform with attention injection to denoise $\boldsymbol{x}_l$ to an enhanced latent $\tilde{\boldsymbol{x}}_t \sim \tilde{p}_{\phi}(\tilde{\boldsymbol{x}}_t)$. Then, we detach $\tilde{\boldsymbol{x}}_t$ from the computation graph to make it an optimization target, so that the gradients of Eq. 5 will backpropagate to the DMTet parameter $\theta$ only through $\boldsymbol{x}_t$. By applying L2 loss on the noisy state $\boldsymbol{x}_t$ from the inversion process and the corresponding enhanced state $\tilde{\boldsymbol{x}}_t$ from the sampling process, the distribution of noisy novel views can be optimized to approach the distribution of their enhanced version.

However, in Eq. 4, unlike regular forward process $q$ where the gradients of $\boldsymbol{x}_t = \alpha_t \boldsymbol{x}_0 + \sigma_t \epsilon$ can be efficiently calculated, DDIM inversion $q_{\phi}$ requires multiple forward-pass of U-Net $\epsilon_{\phi}$, whose gradient is expensive to compute. We therefore turn to a more computationally efficient way to compute gradients of $\boldsymbol{x}_t$. Instead of constructing $\boldsymbol{x}_t$ by adding noise to $\boldsymbol{x}_0$ step by step using DDIM inversion, we can use the deterministic noise $\tilde{\epsilon}_t$ predicted from $\tilde{\boldsymbol{x}}_t$ in the DDIM sampling process to construct noisy states for the rendered novel view, so that the resulted $\boldsymbol{x}_t = \alpha_t \boldsymbol{x}_0 + \sigma_t \tilde{\epsilon}_t$ will have the same noisy level with $\tilde{\boldsymbol{x}}_t$. By this means, the gradients of RGSD loss in Eq. 5 can be efficiently computed and backpropagated to DMTet parameters $\theta$.

We name it Reference-Guided State Distillation (RGSD) loss, as it operates on deterministic noisy states and helps to distill from the DDIM sampling process of our reference-guided novel view enhancement pipeline. We provide a summarized algorithm of RGSD in Algorithm 1. In Appendix. A.2, we demonstrate that our RGSD loss effectively resolves the issues of oversaturation and inconsistent views caused by SDS loss. Please also refer to Appendix. A.1 for more implementation details of the two training stages.

## 4 EXPERIMENTS

### 4.1 ZERO-SHOT NOVEL VIEW SYNTHESIS

**Baseline.** We use Zero-1-to-3-XL (Liu et al., 2023b; Deitke et al., 2023) as the baseline method to assess our novel view enhancement pipeline. Zero-1-to-3 (Liu et al., 2023b) is a diffusion model-based zero-shot novel view synthesis method. By fine-tuning stable diffusion (Rombach et al., 2022) on the Objaverse (Deitke et al., 2023) 3D dataset, it allows for explicit control over the generation of novel views through relative camera pose. By training on a larger 3D dataset Objaverse-XL (Deitke et al., 2023), Zero-1-to-3-XL has achieved state-of-the-art results in zero-shot novel view synthesis.

**Comparison on single view dataset.** We compare our method with the baseline using 400 images, including challenging real-world images and realistic images generated by a T2I model (DF). Fig. 4 presents the qualitative results. We found that although the images generated by Zero-1-to-3-XL exhibit reasonable geometry, their textures often lack details and appear to be unreasonable, resulting in poor consistency with the reference image. On the other hand, our enhancement pipeline produces high-fidelity novel views with realistic textures.

Similar to previous research (Tang et al., 2023a; Qian et al., 2023; Xu et al., 2023), we adopt contextual distance (Mechrez et al., 2018) and CLIP-similarity (Radford et al., 2021) to quantitatively measure the consistency between reference image and novel views. To ensure a fair comparison, we mask out the background generated by our method when computing the metrics since Zero-1-to-3

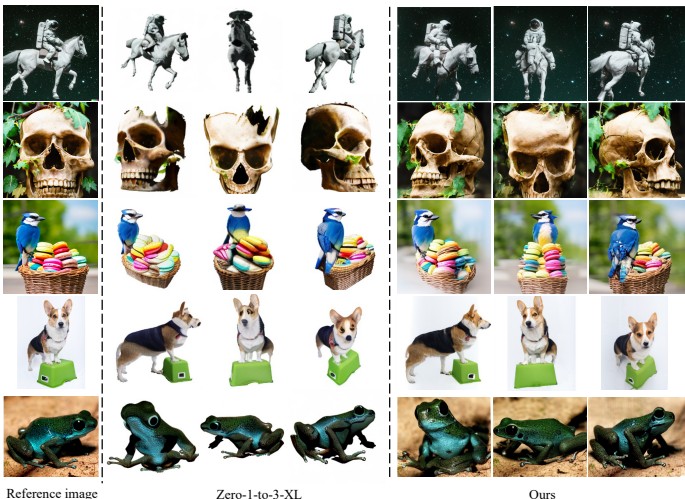

Reference image          Zero-1-to-3-XL                    Ours

Figure 4: Qualitative comparison with Zero-1-to-3-XL on novel view synthesis. It can be found that our method can generate novel views with higher fidelity and consistency.

Table 1: Quantitative comparison with Zero-1-to-3-XL.

|  | Contextual↓ | CLIP↑ |
| --- | --- | --- |
| Zero-1-to-3-XL | 1.94 | 0.89 |
| Ours | **1.60** | **0.93** |

Table 2: Quantitative comparison with image-to-3D generation baselines.

|  | LPIPS↓ | Contextual↓ | CLIP↑ |
| --- | --- | --- | --- |
| RealFusion | 0.35 | 2.18 | 0.81 |
| Make-It-3D | 0.23 | 1.97 | 0.90 |
| Magic123 | **0.19** | 1.88 | 0.88 |
| Ours | 0.21 | **1.63** | **0.92** |

can not generate images with the background. The results are listed in Tab. 1. The improvements in the evaluation metrics reflect our ability to generate novel views that are more consistent with the reference image.

**Comparison on multi-view dataset.** Following (Liu et al., 2023b), we adopt the Google Scanned Object dataset (Downs et al., 2022) and randomly select 30 objects for qualitative and quantitative evaluation. The results are reported in Appendix. A.3.

### 4.2 IMAGE-TO-3D GENERATION

**Baselines.** We compared our image-to-3D generation framework against three state-of-the-art methods: RealFusion (Melas-Kyriazi et al., 2023), Make-It-3D (Tang et al., 2023a) and Magic123 (Qian et al., 2023). RealFusion is a one-stage method that reconstructs NeRF representation from the reference image using L2 reconstruction loss and 2D SDS loss provided by the T2I model. Make-It-3D is a two-stage method that uses 2D SDS loss. It leverages point cloud representation in the second stage for training at higher resolution. Magic123 is also a two-stage method that uses both 2D SDS loss and 3D SDS loss provided by Zero-1-to-3 to balance between geometry and texture quality. We compare all the baselines using their official code.

**Comparison on single view dataset.** We firstly conducted quantitative and qualitative comparisons against baseline methods on realistic single view images. Fig. 5 displays the qualitative comparison results between our method and the baselines. For each input image, we showcase two novel views. For a more comprehensive comparison, please refer to the supplementary videos. We also present a comparison of the normal map optimized by Magic123 and our method. It can be observed that, under the viewpoint that deviates significantly from the reference image, all the baseline methods fail to generate reasonable textures, while our method can maintain the same texture details as the reference image, which greatly improves the fidelity of the generated 3D assets.

For quantitative evaluation, we compute metrics on a total of 19 images. Except for adopting CLIP-similarity and contextual distance for evaluating view consistency, we also use LPIPS (Zhang et al.,

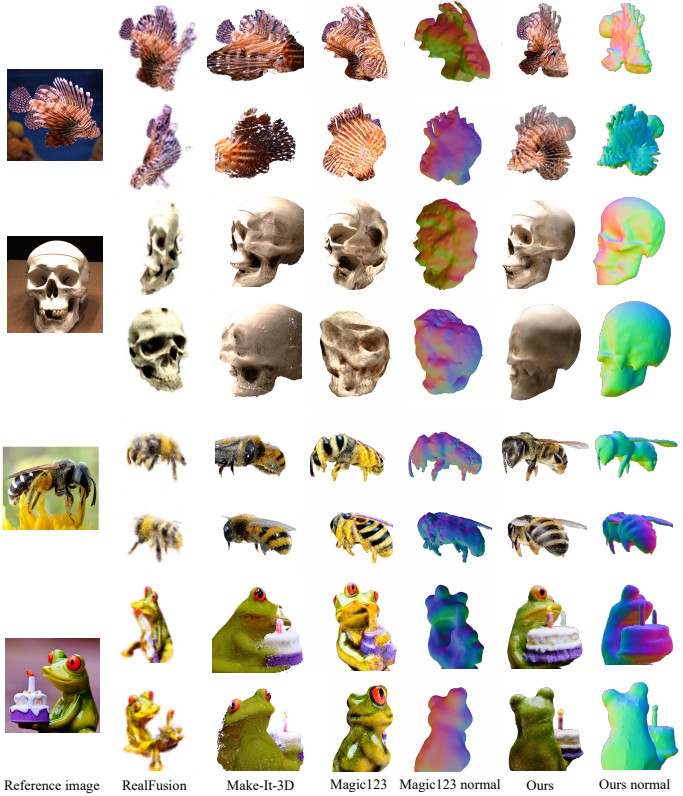

Reference image RealFusion Make-It-3D Magic123 Magic123 normal Ours Ours normal

Figure 5: Qualitative comparison with baselines. Our method outperforms baselines in maintaining texture details under significantly deviating viewpoints.

2018) to evaluate the quality of reference view reconstruction. The results are shown in Tab. 2. The results on CLIP-similarity and contextual distance indicate that our method can generate 3D objects with better 3D consistency. It is worth noting that although Magic123 achieved better results in reference view reconstruction than us by using a larger reference view reconstruction loss, there is a significant inconsistency between their reference view and novel view. This inconsistency is particularly evident at the boundaries between invisible and occluded areas, resulting in noticeable seams. Our method, on the other hand, does not exhibit this issue.

**Comparison on multi-view dataset.** We adopt the Google Scanned Object dataset (Downs et al., 2022) and use 10 objects for multi-view evaluation. The results are reported in Appendix. A.3.

## 5 CONCLUSION AND DISCUSSION

**Conclusion.** We introduce HiFi-123, a method that can be applied for generating high-fidelity novel views in a zero-shot manner as well as high-quality 3D models. Our approach has two key contributions. Firstly, we propose a reference-guided novel view enhancement pipeline, which narrows the quality gap between synthesized and reference views in zero-shot novel view synthesis. Based on this pipeline, we further proposed an RGSD loss to supervise and optimize 3D representations, resulting in highly realistic 3D assets.

**Limitations.** Similar to all optimization-based 3D generation techniques, our image-to-3D framework requires long optimization times. Additionally, Our novel view enhancement pipeline relies on a pre-trained depth-conditioned image diffusion model, which inherently bounds its performance to the capabilities of such models. Moreover, the enhancement process currently requires a coarse novel view to provide an initial depth structure. This makes it more apt as a plug-and-play module for existing zero-shot novel view synthesis methods. Pursuing a pure standalone approach for high-fidelity novel view synthesis remains a promising direction for future research.

## 6 Reproducibility Statement

To ensure the reproducibility of HiFi-123: 1) We will release our code to the community. 2) We will release the reference images we used in experiments. 3) We provide additional comparison videos in the supplementary material. (4) We provide additional implementation details in the appendix.

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

# A   APPENDIX

## A.1   IMPLEMENTATION DETAILS

**Training details.** We train the coarse stage for 5000 iterations using DeepFloyd (DF) guidance, the guidance scale is set to 7.5 and we sample time steps from $t \sim \mathcal{U}(0.2, 0.6)$. After the coarse stage, we train the refine stage for 1000 iterations using our proposed RGSD loss with depth-conditioned SD (SDD) guidance, the guidance scale is set to 7.5 and we sample time steps $t \sim \mathcal{U}(0.0, 0.6)$. We conduct experiments on a 40G A100 GPU, the coarse stage training takes approximately 0.5 hour and the refine stage training takes about 1 hour.

**Coarse Stage NeRF Reconstruction.** In the coarse stage, we adopt instant-ngp(iNGP) (Müller et al., 2022) based NeRF (Mildenhall et al., 2021) as the 3D representation and optimize it with pixel-level image reconstruction loss. Additionally, similar to dreamfusion (Poole et al., 2022), we

---

**Algorithm 1** RGSD loss

---

**Input:** Depth-conditioned SD model $\epsilon_\phi$, reference image, 3D model with parameter $\theta$, attention injection start step $t = l$, learning rate $\eta$.

1: **while** not converged **do**
2:     Randomly sample camera pose $c$ and render $\boldsymbol{x}_0 = g(\theta, c)$
3:     Randomly sample time step $t = \tau$
4:     **for** $t \in [1, l]$ **do**
5:         $\boldsymbol{x}_t = (\alpha_t/\alpha_{t-1})(\boldsymbol{x}_{t-1} - \sigma_{t-1}\epsilon_\phi) + \sigma_t\epsilon_\phi$ {DDIM inversion}
6:     **end for**
7:     **for** $t \in [l, \tau + 1]$ **do**
8:         $\tilde{\boldsymbol{x}}_{t-1} = (\alpha_{t-1}/\alpha_t)(\tilde{\boldsymbol{x}}_t - \sigma_t\tilde{\epsilon}_t) + \sigma_{t-1}\tilde{\epsilon}_t$ {DDIM sampling with attention injection}
9:     **end for**
10:    Construct $\boldsymbol{x}_\tau = \alpha_\tau\boldsymbol{x}_0 + \sigma_\tau\tilde{\epsilon}_\tau$
11:    $\theta \leftarrow \theta - \eta\nabla_\theta\mathbb{E}[\|\boldsymbol{x}_\tau - \tilde{\boldsymbol{x}}_\tau\|_2^2]$
12: **end while**
13: **return**

---

employ SDS loss to encourage semantically plausible results that comply with the text prompts. The chosen iNGP has a 16-level hash encoding of size $2^{19}$ and entry dim 2, and its base resolution is set to 16 with a two-layer MLP with 64 hidden channels to decode density and color.

We also penalize the foreground mask differences between the rendered opacity map and the segmentation map of the reference image. To further regularize the object geometry, we incorporate a reference view depth loss (Tang et al., 2023a) and normal consistency loss (Poole et al., 2022). We train the coarse NeRF at image resolution 128.

**Refine Stage DMTet Reconstruction.** During the refine stage, we choose DMTet(Shen et al., 2021b) as the 3D representation. DMTet is a hybrid SDF-Mesh 3D representation comprising deformable tetrahedral grid $(V_T, T)$ which is capable of differential rendering and explicit high-resolution shape modeling. The deformation vector is initialized to $0$ and SDF is initialized by converting the coarse stage density field.

We train the refine stage at the image resolution of $512$. For the texture field, we employ an iNGP with the same setting as the aforementioned coarse NeRF. The novel view results can be rendered by a differentiable rasterizer which rasterizes extracted mesh from DMTet and the texture field that gets a 3D intersection from the rasterizer as input. During training, except for utilizing reference view L2 loss and RGSD loss to optimize the SDF, deformation vector and texture field, we additionally leverage two enhanced non-overlapping novel views as pseudo ground truth and apply L2 loss in the pixel space for producing sharper textures and training stability.

**Camera Settings.** During the coarse stage optimization, we sample the predefined reference view and a random view in each iteration. During the fine stage optimization, we sample uniformly one from the combined image set comprised of the selected enhanced views and reference views. We also sample a random view for SDS loss back-propagation in each iteration. For the random sampling, the elevation angles in uniformly sampled from [-10, 60], and the azimuth angle is uniformly sampled from [-180, 180].

## A.2   Ablation study

**Depth-based DDIM inversion**. Our novel view enhancement pipeline is built on the discovery that performing DDIM inversion on an input image using a depth-conditioned diffusion model (SDD) can significantly improve the reconstruction quality. As shown in Fig. 6, compared with regular DDIM inversion, depth-based DDIM inversion can significantly improve the reconstruction quality of the input images, even outperforming the optimization-based textual inversion (Gal et al., 2022b), and is comparable to Null-text inversion (Mokady et al., 2023) which is also optimization-based. This enables us to obtain an accurate representation of the input image and adapt it to high-fidelity novel-view synthesis in a zero-shot manner.

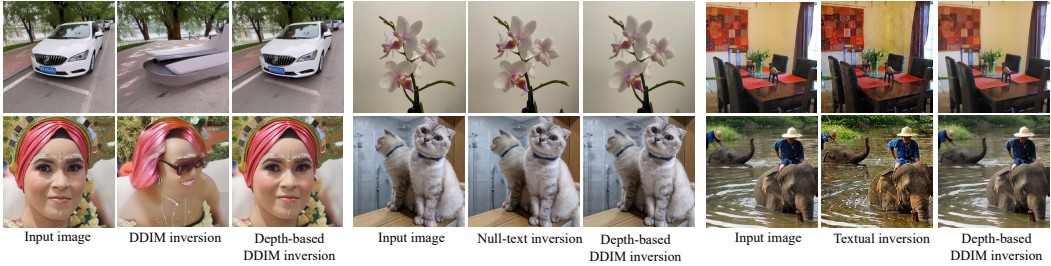

Figure 6: Comparison between depth-based DDIM inversion and other inversion approaches.

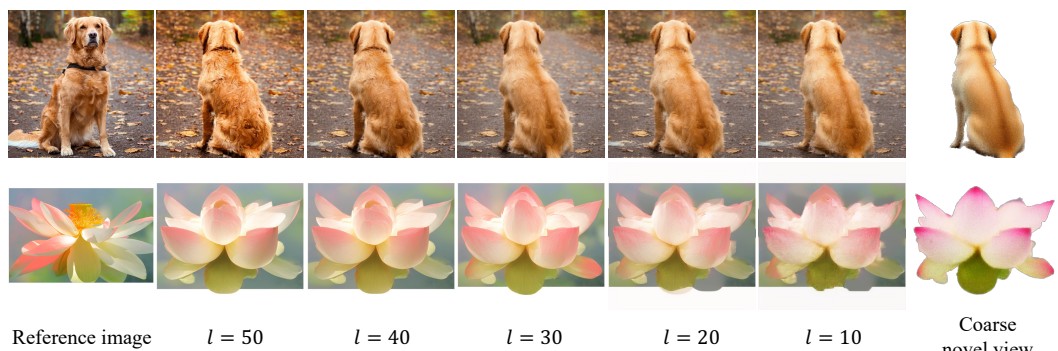

Figure 7: Ablation on time steps

**Ablation on inversion time steps.** As introduced in Section. 3.2, we perform $t = l$ steps' DDIM inversion to invert the reference image and coarse novel view into noisy latent, then perform DDIM sampling on it with attention injection to transfer textures. The impact of different inversion time steps $l$ is shown in Fig. 7. We use the commonly used 50 step's DDIM sampling and inversion in the experiment. It can be observed that as $l$ increases, the texture of the enhanced image approaches that of the reference image more closely, but may introduce geometry change. We use $l = 30$ in image-to-3D generation and use $l = 50$ in novel view synthesis.

**Ablation on depth condition and attention injection in novel view enhancement.** There are two key components in our proposed reference-guided novel view enhancement pipeline: the depth-based DDIM inversion and the attention injection. We qualitatively validate this design choice in Fig. 8. Given a reference image and a rendered coarse novel view, performing novel view enhancement without the depth-conditioned SD model (SDD) leads to enhanced results that are inconsistent with the reference image. The reason lies in that the regular DDIM inversion cannot precisely reconstruct the reference image (see Fig. 6), thus failing to transfer fine textures of reference image to the coarse novel view. Further, directly using the depth-conditioned SD model (SDD) without attention injection to enhance the coarse novel views also results in images inconsistent with the reference. With depth-based DDIM inversion that capture fine details of the reference image and attention injection that transfer fine textures to the coarse novel views, our enhancement pipeline can produce enhanced images consistent with the reference, as shown in the last column.

**Comparison of utilizing RGSD loss and other losses in the refine stage.** Based on our novel view enhancement pipeline, we propose an RGSD loss for refining textures of a coarse 3D representation. Here, we performed an ablation study on using different losses in the refine stage of our image-to-3D generation method. As shown in Fig. 9, the coarse novel views rendered from the coarse stage are of low quality with smooth textures. In Fig. 9. (a), refining with vanilla SDS loss fails to generate colors consistent with the reference image due to the ambiguity of the text prompt. To ensure high fidelity in novel views, we incorporate our proposed reference-guided novel view enhancement pipeline into the refine stage. A naive approach would be to use our enhancement pipeline to generate enhanced images for random camera views and apply reconstruction loss between rendered images and enhanced images, namely using pixel loss. However, Fig. 9. (b) shows even slight inconsistencies in

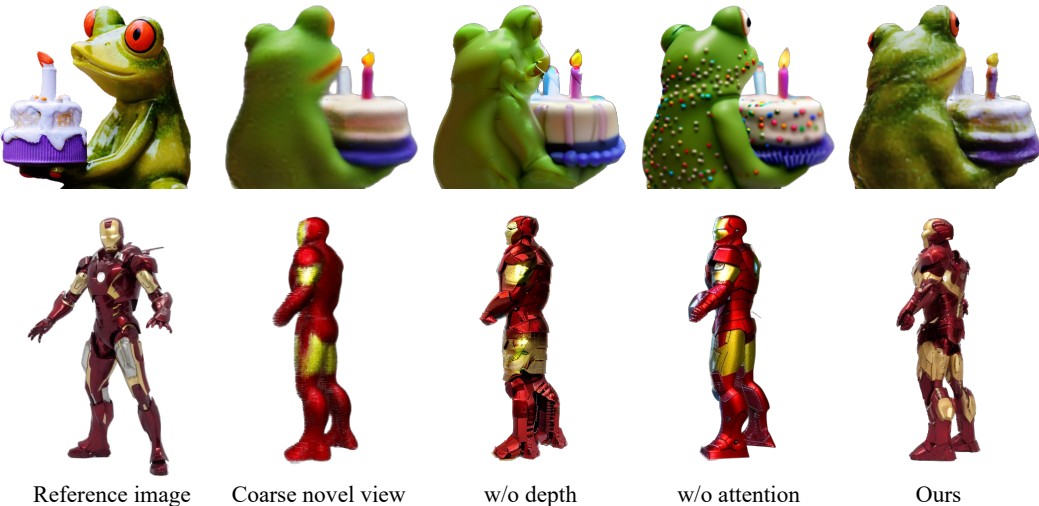

Reference image     Coarse novel view     w/o depth     w/o attention     Ours

Figure 8: Ablation on depth condition and attention injection.

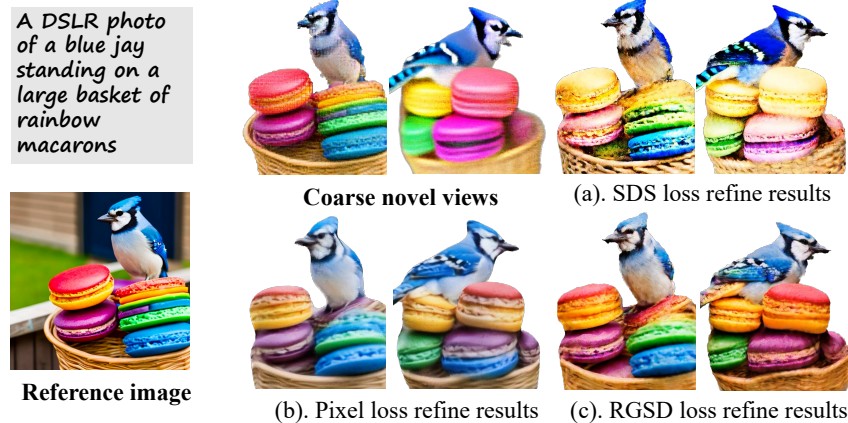

Figure 9: Comparison of using different losses in the refine stage.

the overlapping areas between enhanced novel views will accumulate and lead to blurry optimized results. In contrast, as shown in Fig. 9. (c), our RGSD loss performs state distillation in the latent space of the stable diffusion model, enabling faithfully generating consistent appearance with the reference image.

### A.3 MORE COMPARISONS.

**Novel view synthesis comparison on multi-view dataset.** We adopt the commonly used PSNR, SSIM (Wang et al., 2004) and LPIPS (Zhang et al., 2018) to quantitatively evaluate the novel view synthesis quality, the results are shown in Tab. 3. Our method achieves better performance in all the metrics, indicating that the reference-guided novel view enhancement pipeline helps to improve fidelity of the novel views. Qualitative results are displayed in Fig. 10, showing that our method can generate textures that are more consistent with the reference image.

**Image to 3D generation comparison on multi-view dataset.** For image-to-3D generation evaluation with ground truth data, we adopt PSNR, SSIM (Wang et al., 2004) and LPIPS (Zhang et al., 2018) to quantitatively evaluate the novel view synthesis quality, and utilize the commonly used Chamfer Distances and Volume IoU to evaluate the generated geometry. As shown in Fig. 11, our method produces more reasonable texture and material information that are consistent with the ref-

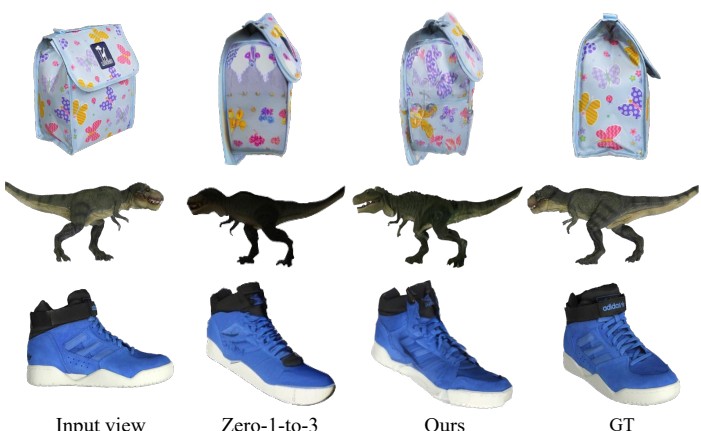

Table 3: Quantitative comparison with zero-1-to-3 on GSO dataset.

|        | Zero-1-to-3 | Ours      |
| ------ | ----------- | --------- |
| PSNR↑  | 34.58       | **36.89** |
| SSIM↑  | 0.763       | **0.795** |
| LPIPS↓ | 0.231       | **0.214** |

Input view     Zero-1-to-3     Ours     GT

Figure 10: Qualitative comparison with Zero-1-to-3 on GSO dataset.

Table 4: Quantitative comparison with image-to-3D generation baselines on GSO dataset.

|            | PSNR↑     | SSIM↑     | LPIPS↓    | Chamfer Dist↓ | Volume IoU↓ |
| ---------- | --------- | --------- | --------- | ------------- | ----------- |
| Make-it-3D | 22.38     | 0.734     | 0.251     | 0.086         | 0.368       |
| Magic123   | 30.16     | **0.789** | 0.237     | 0.066         | 0.461       |
| Ours       | **32.05** | 0.785     | **0.221** | **0.055**     | **0.493**   |

erence image. Tab. 4 shows the quantitative results, which indicates that our method is capable of generating 3D contents with better geometry and textures comparing with the strong baselines.

**Comparison with syncdreamer.** We qualatitively compare with Syncdreamer (Liu et al., 2023c), the reslts are shown in Fig. 12. Our method can generated novel views more consistent with the referene image.

## A.4 MORE RESULTS.

We shown more results of the 3D contents generated by our method. As shown in Fig. 13, our method can produce high fidelity 3D contents with consistent appearances. The corresponding coarse novel view are displayed in the last tow columns, which demonstrates the effectiveness of the refine stage.

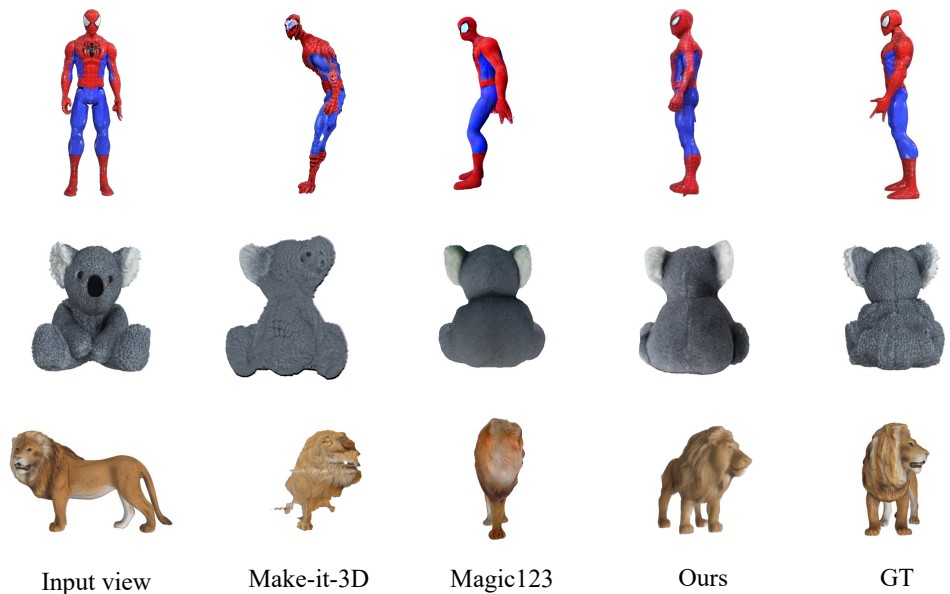

| Input view | Make-it-3D | Magic123 | Ours | GT |

Figure 11: Qualitative comparison with image-to-3D generation baselines on GSO dataset.

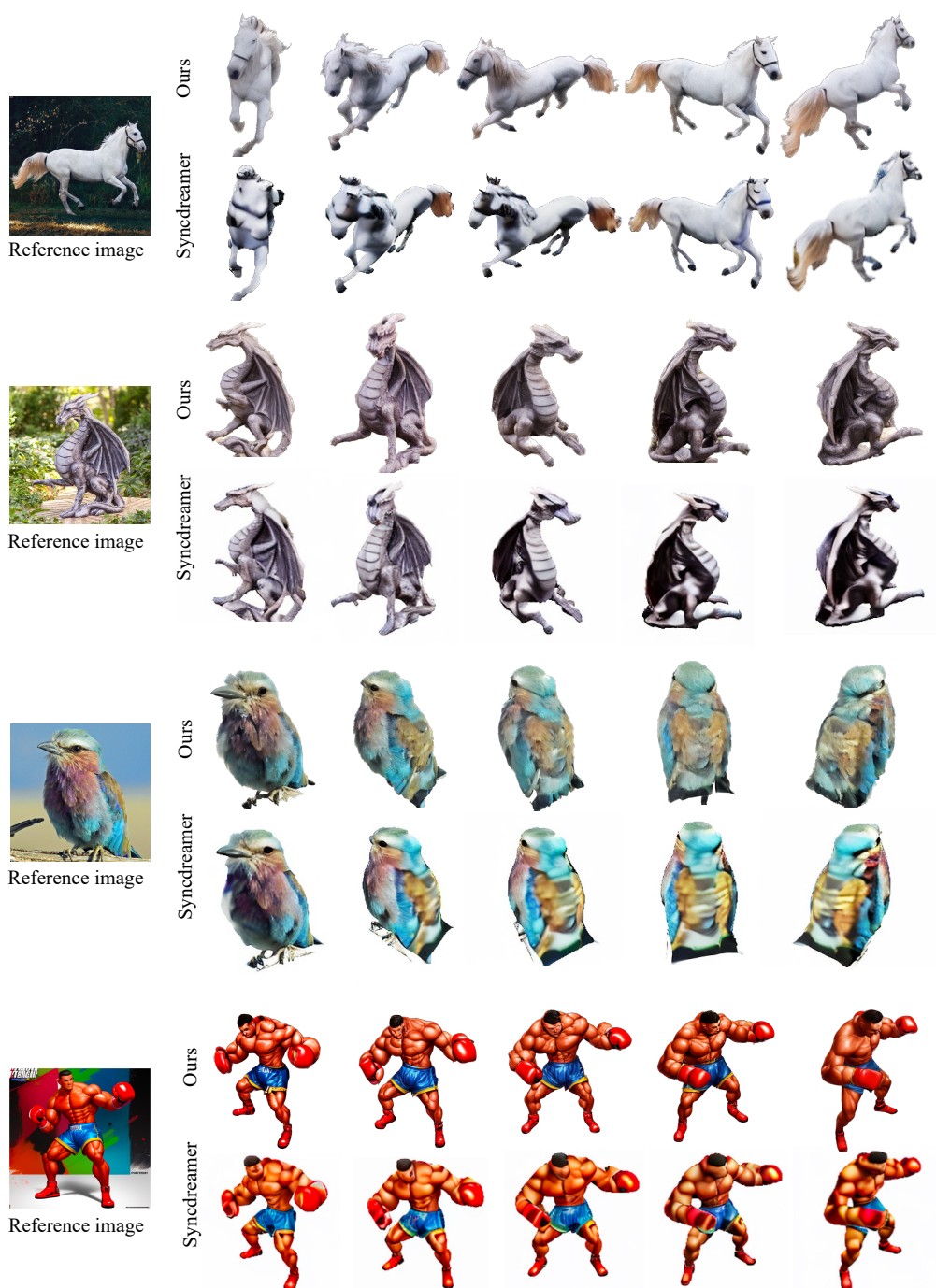

Figure 12: Qualitative comparison with Syncdreamer. It can be found that our method can generate novel views with higher fidelity according to the reference image.

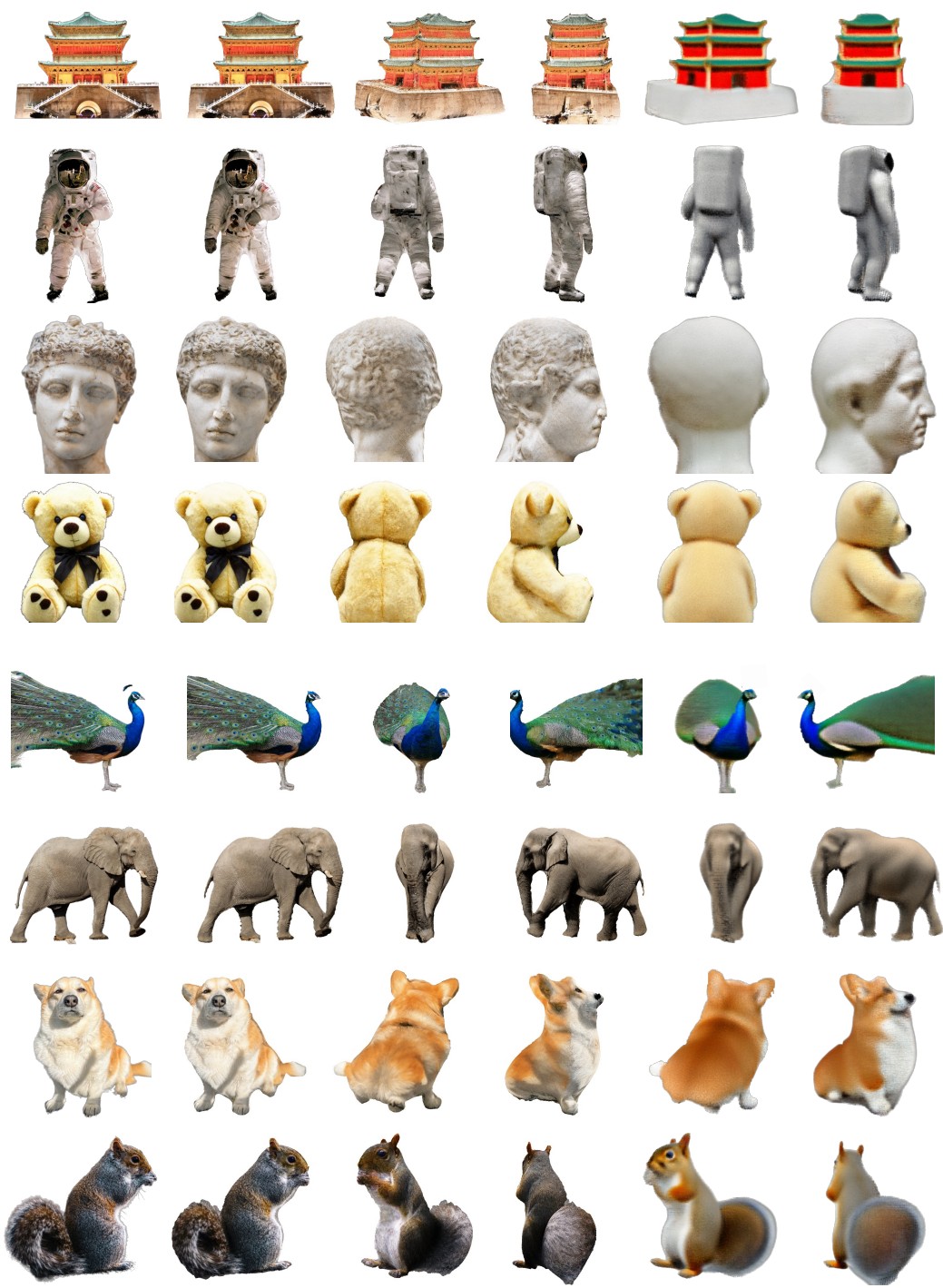

Figure 13: Additional image-to-3D generation results by our method. We show the corresponding coarse novel view in the last tow columns.

