# OpenReview forum: "HiFi-123: Towards High-fidelity One Image to 3D Content Generation"
_ICLR.cc/2024/Conference — ICLR 2024 Conference Withdrawn Submission_

### Official Review · Reviewer_WKaM · 2023-10-30

**Soundness:** 2 fair
**Presentation:** 2 fair
**Contribution:** 2 fair
**Rating:** 3
**Confidence:** 5

**Summary:**

The authors propose a method for image-to-3D generation. Specifically, they suggest incorporating depth images during DDIM inversion and sampling to generate view-consistent novel view images. The enhanced novel view images are then used to compute a variant version of SDS loss, i.e., the RGSD loss, to optimize the 3D representation. The results appear to outperform the baselined methods used.

**Strengths:**

The illustration is generally clear, although some sections could be further improved.

Both the visual and quantitative results surpass those of the proposed baselines.

**Weaknesses:**

Mistakes:

1) In Fig.1, is the inconsistency between the depth map and the novel view image in the bottom line a mistake?

Results:

1)  More diverse rendering results are expected  such as buildings and human bodies, as are in other prior works.

2) Flickering issues are observable in the videos. An explanation or analysis is expected.

Evaluations:

1) I doubt about the effectiveness of reference view reconstruction for evaluation. One may achieve excellent results on the reference view but less consistency on novel views. Using held-out data for novel view reconstruction might be more informative. Additionally, 3D evaluations are encouraged, e.g., utilizing Chamfer distance and Volume IoU.

2) A more comprehensive ablation study is recommended. The current ablation results are only displayed in Fig.5, examining different losses. In addition, ablation for other design choices is needed. For instance, what are the results for novel views without utilizing depth-inserted inversion, and how much does the proposed inversion method enhance its quality?

Concurrent works:

1) In addition to the comparison with zero-1-to-3, the paper also mentions other concurrent works, such as MVDream and SyncDreamer. Is it possible to also make comparisons with these in some way?

Miscs:

1) Implementation details are absent, including training time, GPU resource consumption, and hyper-parameters of SDS, such as timestep scheduling and guidance scale.

2) Section 3.1 could benefit from a rewrite with clearer organization. Currently, it seems somewhat distracted by the variants of notations, even for the same variable, e.g., x_t (\theta, \phi) and x_t (\theta), and unclear definitions of \tilde{x}\_t and its two forms of distributions: \tilde{p}\_\phi (\tilde{x}\_t) and  \tilde{p}\_\phi (\tilde{x}\_t; y, m, t).

3) Additionally, an algorithm could be included to aid in understanding the training process.

**Questions:**

See weaknesses above.

**Details Of Ethics Concerns:**

Reconstructing high-fidelity human identities or other IP-sensitive assets may raise ethical concerns

---

> ### Author Response · Authors · 2023-11-16
> **Response to Reviewer WKaM**
>
> We thank the reviewer's advice and responses as follows. Please also check our revised paper for detailed results.
>
> **Q1. In Fig.1, is the inconsistency between the depth map and the novel view image in the bottom line a mistake.**
>
> **A1.** Thanks for pointing out the mistake, we corrected it in the revised paper.
>
> **Q2.More diverse rendering results are expected such as buildings and human bodies, as are in other prior works.**
>
> **A2.** Thanks for the suggestion. We show more results in Figure 13 in the revised paper.
>
> **Q3.Flickering issues are observable in the videos. An explanation or analysis is expected.**
>
> **A3.** We found this is caused by using L2 loss to optimize the DMTet. This issue can be significantly mitigated by additionlly using LPIPS loss, examples are in the newly uploaded supplementary.
>
> **Q4.Using held-out data for novel view reconstruction might be more informative. Additionally, 3D evaluations are encouraged, e.g., utilizing Chamfer distance and Volume IoU.**
>
> **A4.** Thanks for the advice, in the revised paper, we add comparisons on the GSO dataset with ground truth shape and novel views, and discuss the experiments in **Appendix A.3**
>
> **Q5. A more comprehensive ablation study is recommended.**
>
> **A5.** Thanks for the advice, in the revised paper, we add ablation studies in **Appendix A.2**, including the ablation on the results for novel views without utilizing depth-inserted inversion.
>
> **Q6. In addition to the comparison with zero-1-to-3, the paper also mentions other concurrent works, such as MVDream and SyncDreamer. Is it possible to also make comparisons with these in some way?**
>
> **A6.** MVDream is a text-to-image model, which is not designed for zero-shot novel view synthesis. We qualitatively compare with Syndreamer on zero-shot novel view synthesis, results are displayed in Figure 12.
>
> **Q7. Implementation details are absent, including training time, GPU resource consumption, and hyper-parameters of SDS, such as timestep scheduling and guidance scale.**
>
> **A7.** Thanks for pointing out this, we provide more implementation details in **Appendix A.1**, inculding training details in the revised paper.
>
> **Q8.Section 3.1 could benefit from a rewrite with clearer organization.**
>
> **A8.** Thanks for the suggestion, we improved the organizatin and notations in Section 3.1 and 3.3 in the revised paper.
>
> **Q8. Additionally, an algorithm could be included to aid in understanding the training process.**
>
> **A8.** Thanks for the advice, we incude an algorithm of the RGSD loss in the revised paper.

---

### Official Review · Reviewer_jpP8 · 2023-10-31

**Soundness:** 3 good
**Presentation:** 3 good
**Contribution:** 3 good
**Rating:** 5
**Confidence:** 4

**Summary:**

This paper introduces HiFi-123, an approach for achieving high-fidelity novel view synthesis and generating image-to-3D content with multi-view consistency. The authors present a technique called reference-based novel view enhancement, aimed at bridging the texture quality gap between the synthesized novel views and the reference view. Building upon it, they propose a reference-guided state distillation loss for 3D generation. They also propose to exploit a pretrained depth conditioned diffusion model as the base model for distillation. The application of their method leads to improvements in the quality of both generated novel views and 3D content.

**Strengths:**

* The paper is well written and easy to follow;
* The proposed method improves the performance of image-to-3D content creation compared to prior methods;

**Weaknesses:**

The ablation study is not sufficient:
* The extent to which the improvement is attributable to the depth conditioned stable diffusion model or the attention injection remains unclear；
* The extent to which the novel view enhancement pipeline will be affected by the quality of the rendered coarse view depth map remains unclear;
* Can you also present the result of the generated 3D content after the coarse stage, so that we can see the improvements by the refine stage? Is it in Figure 5, i.e. the SDS loss result? If yes, can you present more results from additional text prompts?

Without those ablation studies, the reviewer is unable to properly justify the contributions of the work, i.e. I am uncertain if the performance improvements compared to prior methods are attributable more by the used DeepFLoyd-IF model (which could be trivially adopted by prior methods) or the proposed enhancement technique. Please address them in the rebuttal.

**Questions:**

* In Eq5, why do not use the recovered x_0 to compute the loss against the rendered image directly? Would it lead to similar or better performance? Could the authors present the comparison results?

* In Figure 1, do you need to retrain the bottom network, since the input coarse novel view & depth map are usually in low quality?

---

> ### Author Response · Authors · 2023-11-16
> **Response to Reviewer jpP8**
>
> We thank the reviewer's comments and make responses as follows. Please also check our revised paper for detailed results.
>
> **Q1.The extent to which the improvement is attributable to the depth conditioned stable diffusion model or the attention injection remains unclear.**
>
> **A1.** We add an ablation on the role of depth condition and attention injection in **Appendix A.2**. The results in **Figure 8** shows both the two components are important to our enhancement pipeline.
>
> **Q2.The extent to which the novel view enhancement pipeline will be affected by the quality of the rendered coarse view depth map remains unclear**
>
> **A2.** Since the depth-conditioned SD model is trained on depth maps estimated by Midas, We therefore use pretrained Mida depth estimater to generate depth map for coarse novel views, and use foreground mask to get the object region. The conditioned depth map acts as a shape guidance, and the  depth values do not need to be accurate.
>
> **Q3. Can you also present the result of the generated 3D content after the coarse stage, so that we can see the improvements by the refine stage? Is it in Figure 5, i.e. the SDS loss result? If yes, can you present more results from additional text prompts?**
>
> **A3.** Thanks for the advice. In the revised paper, we present the result of the generated 3D content after the coarse stage in Figure 13. The SDS loss results in Figure 5 (now Figure 9) shows refine results produced by SDS, it is not the coarse result, we have improved Figure9 to make it clearer.
>
> **Q4. In Eq5, why do not use the recovered x_0 to compute the loss against the rendered image directly? Would it lead to similar or better performance? Could the authors present the comparison results?**
>
> **A4.** As shown in Figure 9. (b) in the revised paper, this will lead to blurry textures. Analysis are in **Appendix. A.2, Comparison of utilizing RGSD loss and other losses in the refine stage**.
>
> **Q5. In Figure 1, do you need to retrain the bottom network, since the input coarse novel view & depth map are usually in low quality?**
>
> **A5.** We do not need to retrain the bottom network, we directly use the officially released pretrain model. The conditioned depth map acts as a shape guidance, the depth values do not need to be accurate and the depth-conditinoed model is robust to handle different input quality.

---

### Official Review · Reviewer_gkLK · 2023-10-31

**Soundness:** 3 good
**Presentation:** 3 good
**Contribution:** 3 good
**Rating:** 8
**Confidence:** 4

**Summary:**

The paper proposes a novel image-to-3D method that involves a combination of techniques to improve the quality and consistency of the result 3D shape. The proposed method consists of two stages:
* The coarse stage optimizes an Instant NGP with depth, surface normal, and RGB reconstruction loss on the reference view, as well as SDS loss on random views using a text-to-image diffusion model. The depth and surface normal are estimated using off-the-shelf models, and the text-to-image model is pretrained.
* The fine stage is switched to DMTet for higher resolution. It is optimized with reconstruction loss on the reference view, and the newly proposed reference-guided state distillation (RGSD) loss based on a depth-to-image diffusion model.

The key contribution of the paper is the RGSD loss, which involves DDIM inverting the ground truth reference image and novel view rendering to a common time step, then injecting the attention weights from the reference image to the novel view while denoising both images. This will result in an enhanced novel view image that contains similar details compares to the reference view. This enhanced image is then used to supervise the novel view rendering.

The 3D shapes produced by the proposed method is very faithful to the reference image while looking good from all directions, as verified both visually and through automated evaluations including LPIPS distance, contextual distance, and CLIP similarity.

**Strengths:**

* The use of DDIM inversion + attention injection, in the context of image-to-3D, is not only quite novel, but also performing very well -- its effects are ablated in the appendix.
* The proposed method produces 3D shapes that are significantly more consistent with the reference view while having better visual quality when looked from unseen directions. This can be seen both numerically from the overall better evaluation metrics, as well as empirically from visual results.
* The method does not involve training or finetune any large foundation models -- this will save computation and make the method more accessible.

**Weaknesses:**

* The key technique used in the reference-guided novel view enhancement method proposed in the paper is not completely new. It has been incorporated in diffusion-based video generation [Wu et al. 2022, Qi et al. 2023] and image editing [Cao et al. 2023].
* The majority of the contributions of the paper focuse on the "fine" stage, while the "coarse" stage still relies on SDS and reconstruction loss. The fine stage will likely not be able to recover from the mistakes in the coarse stage. It will also be good to show results from the coarse stage so that the effect of the proposed methods can be better appreciated.
* A large number of pretrained models are involved during inference, which can be complicated:
  * Depth estimation model
  * Surface normal estimation model
  * DeepFloyd-IF text-to-image model
  * stable-diffusion-2-depth depth+text-to-image model

**Questions:**

* Could you elaborate more on the "Figure 5: Ablation on RGSD loss" in the appendix? It is especially unclear how the "Pixel loss" result is obtained.
* How long does it typically take to generate one 3D shape?

---

> ### Author Response · Authors · 2023-11-16
> **Response to Reviewer gkLK**
>
> We appreciate the reviewer's very positive comments and make responses as follows. Please also check our revised paper for detailed results.
>
> **Q1. It will also be good to show results from the coarse stage so that the effect of the proposed methods can be better appreciated.**
>
> **A1.** Thanks for your advice, in the revised paper, we show more results of the coarse stage in **Figure 13**.
>
> **Q2. Could you elaborate more on the "Figure 5: Ablation on RGSD loss" in the appendix? It is especially unclear how the "Pixel loss" result is obtained.**
>
> **A2.** We improved Figure 5 (now Figure 9) in the revised paper, and elaborate more on  it in **Appebdix A.2, Comparison of utilizing RGSD loss and other losses in the refine stage**.  Please check the revised paper for details.
>
> **Q3.How long does it typically take to generate one 3D shape?**
>
> **A3.** It takes about 1.5 hour on a 40G A100 GPU, we add the training details in the revised paper, in **Appendix A.1, Training details**

---

### Official Review · Reviewer_vttR · 2023-11-02

**Soundness:** 3 good
**Presentation:** 3 good
**Contribution:** 3 good
**Rating:** 5
**Confidence:** 5

**Summary:**

The paper proposes a novel method to improve the fidelity of 3D generation from a single image based on a diffusion model. The main contribution of the paper is combining DDIM inversion with Zero123. A novel view synthesized by Zero123 can be enhanced by performing DDIM inversion and sampling in terms of quality and consistency.

**Strengths:**

1. The method seems effective. Both qualitative and quantitative experiments are conducted to demonstrate the effectiveness of the proposed method
2. The presentation of the method is clear and easy to understand
3. The paper is mostly self-contained. Relavent prior works are cited and necessary preliminary concepts are introduced

**Weaknesses:**

The quantitative experiments are relatively weak for several reasons:

a. lack of evaluation of 3D generated geometry: the proposed method is claimed to enhance the fidelity of 3D generation. However, there's no quantitative evaluation on 3D geometry or texture. There are plenty of datasets such as GSO, RTMV, CO3D where such evaluation can be done in a standardized manner. I think this will be necessary in showing the effectiveness of the approach.

b. lack of evaluation of 3D consistency: one of the claimed contribution of the paper is the improved 3D consistency in generated images compared to zero123. However, this is not evaluated quantitatively. I understand that the evaluation of consistency may not be easy but some approximate metrics such as how well a pair of novel view images satisfy the epipolar constraints can be used.

c. lack of evaluation on novel view synthesis with 3D ground truth: the evaluation focuses on using 2D images and evaluates by comparing the CLIP similarity with the reference input view. This is very problematic because under such metric, an optimal model will be one that always reconstructs the input image. I think quantitative evaluation of novel view synthesis has to be done on a multiview dataset and against, these datasets and benchmarks are readily available.

**Questions:**

1. Why does the proposed model produces images with background in figure 3 but images without background in figure 4?
2. An important assumption made in the paper is that the coarse novel view image generated contain the correct global structure of the object and the details of the image are most needed for improvement. I'm not sure if this assumption is correct since sometimes the generated coarse novel view can be wildly off even in global structure for various reasons.

---

> ### Author Response · Authors · 2023-11-16
> **Response to Reviewer vttR**
>
> We appreciate the reviewer's comments and make responses as follows. Please also check our revised paper for detailed results.
>
> **Q1. lack of evaluation of 3D generated geometry.**
>
> **A1.** Thanks for your advice, in the revised paper, we compare with image-to-3D generation baselines on the GSO dataset in Appendix **A.3**. We use Chamfer Distance and Volume IOU to evluate the generated geometry. Qualitative and quantitative results are in **Table 4** and **Figure 11**.
>
> **Q2. lack of evaluation of 3D consistency.**
>
> **A2.** Thank you for your agreement with the difficulty in consistency evaluation. We found the suggested epipolar evaluation are mostly used for qualitative evaluation, such as in NeRF-W [1]. For quantitative evlation, we think compare with zero123 on multi-view dataset with ground truth novel views can also demonstrated the 3D consistency, we therefore compare with zero123 on GSO dataset. Results are reported in  Appendix **A.3**, **Table 3** and **Figure 10**.
>
> [1]NeRF in the Wild: Neural Radiance Fields for Unconstrained Photo Collections. CVPR2021
>
> **Q3. lack of evaluation on novel view synthesis with 3D ground truth**
>
> **A3.** In the revised paper, we compare with zero123 on GSO dataset, Results are reported in  Appendix **A.3**, **Table 3** and **Figure 10**.
>
> **Q4. Why does the proposed model produces images with background in figure 3 but images without background in figure 4?**
>
> **A4.** Our method has the ability to transfer background from the reference image to novel views. figure3 (now figure4 in the revision) shows the zero-shot novel view synthesis results, we keep the original background. figure4 (now figure5) shows image-to-3D generation reuslts, similar with baselines, the training process requires mask out the background, therefore the generated 3D content are without background.
>
> **Q5. An important assumption made in the paper is that the coarse novel view image generated contain the correct global structure of the object and the details of the image are most needed for improvement.**
>
> **A5.** One the one hand, recent diffusion model-based zero-shot novel view synthesis methods, like zero123, are fine-tuned from stable diffusion models on the Objaverse dataset. This will lead to decline in the image quality, and our method can supplement fine details for these models in a zero-shot manner.
> On the other hand, Image-to-3D generation generated using SDS loss suffers from over-smooth textures and statured colors, our method can also effectively resolve the prolem.
> However, we do admmit that our method will inherrit the wrong coarse geomety, we have discussed this the the **CONCLUSION** section in the original paper.

---

### Author Response · Authors · 2023-11-16
**Overall Response**

Thank you for your valuable comments. We have carefully considered all suggestions and feedback provided by the reviewers and have revised our paper accordingly. The changes made to the paper include:

**1. Correct of the figures**

**2. Provide more implementation details**

**3. Improve Section 3.1 and 3.3 with clearer organization**

**4. Add an algorithm to aid in understanding the training process**

**5. Additional evaluation on the GSO dataset**

**6. Additional ablation studies**

**7. Providing more results and results of the coarse stage**

Please check our revised paper for details. We appreciate your time and effort in reviewing our submission and look forward to your continued feedback. Thank you again for your thoughtful comments

---

### Author Response · Authors · 2023-11-17
**Request for discussion**

Dear reviewers, we would like to express our sincere gratitude again for your diligent review work and invaluable feedback on our submission. We are eager to engage in further discuss with you regarding the revised submission of our work, as we are committed to continuously enhancing its quality. We kindly request your response so that we can have time to address additional questions and concerns. Thank you for your consideration and ongoing support.